# In-situ Deposition of Graphene Oxide Catalyst for Efficient Photoelectrochemical Hydrogen Evolution Reaction Using Atmospheric Plasma

**DOI:** 10.3390/ma13010012

**Published:** 2019-12-18

**Authors:** Khurshed Alam, Yelyn Sim, Ji-Hun Yu, Janani Gnanaprakasam, Hyeonuk Choi, Yujin Chae, Uk Sim, Hoonsung Cho

**Affiliations:** 1Department of Materials Science and Engineering, Chonnam National University, Gwangju 61186, Korea; materialengineer171@gmail.com (K.A.); simyelyn0804@gmail.com (Y.S.); janani.gkovai@gmail.com (J.G.); gkdis4285@gmail.com (H.C.); ghkalwls@gmail.com (Y.C.); 2Center for 3D Printing Materials Research, Korea Institute of Materials Science, Changwon 41508, Korea; jhyu01@kims.re.kr

**Keywords:** photoelectrochemical cell, graphene oxide, hydrogen, atmospheric plasma, hydrophobicity, hydrophilicity

## Abstract

The vacuum deposition method requires high energy and temperature. Hydrophobic reduced graphene oxide (rGO) can be obtained by plasma-enhanced chemical vapor deposition under atmospheric pressure, which shows the hydrophobic surface property. Further, to compare the effect of hydrophobic and the hydrophilic nature of catalysts in the photoelectrochemical cell (PEC), the prepared rGO was additionally treated with plasma that attaches oxygen functional groups effectively to obtain hydrophilic graphene oxide (GO). The hydrogen evolution reaction (HER) electrocatalytic activity of the hydrophobic rGO and hydrophilic GO deposited on the p-type Si wafer was analyzed. Herein, we have proposed a facile way to directly deposit the surface property engineered GO.

## 1. Introduction

Graphene has attracted great attention through decades due to high stability, high surface area, good thermal conductivity, and fast carrier mobility [1]. Along with the numerous advantages, the use of graphene in electrochemical energy application requires high energy to sustain high temperature and pressure under the deposition process and it is difficult to form the bulk system [2,3]. Reduced graphene oxide (rGO) is a promising alternative for bulk production of graphene-like materials. The bottleneck of its commercialization is the control of oxygen functional groups on the surface to engineer its diverse properties, such as electronic structure, optical properties, and surface properties [4,5].

Among various applications of rGO, the most primary usage is the catalysis used in the environment-friendly and cost-effective hydrogen fuel production from renewable energy sources like solar energy. Eventually, hydrogen can be produced by various techniques such as steam reforming of natural gas and electrolysis of water. However, the former technique utilizes fossil fuels and emits greenhouse gases, while the latter hinges on electricity that is relatively costly and less efficient, limiting its large-scale production [6,7]. To bridge the gap between lab-scale production and commercialization, the scientific community has been trying to do electrolysis of water by the photoelectrochemical method, which utilizes sunlight to oxidize or reduce water into oxygen and hydrogen [6].

A photoelectrochemical cell (PEC), which can produce H_2_ by solar energy-driven photocatalysis, is the effectual device for the production of hydrogen. Tailoring an efficient photocathode, which is earth-abundant, cheaper, and durable, stances as a focal point for the development of photoelectrochemical hydrogen production [8,9]. As a result, silicon (Si) has been broadly studied for the photoelectrochemical hydrogen production due to its availability (second abundant element on earth), cost efficiency, ability to absorb light over a wide range of solar spectrum (bandgap of about 1.12 eV), and high theoretical maximum photocurrent density (44 mA cm^−2^) [8,10]. In addition, the bandgap of Si is slightly lower than overall water splitting energetics (1.23 eV), while its conduction band is energetically more negative than the proton reduction potential [11]. Though it seems to be an ideal candidate for the hydrogen evolution reaction (HER), the small band bending between the edge of the valence band and the redox level of electrolyte is the hurdle for the best hydrogen production in liquid electrolyte [12]. Moreover, the durability of Si in the aqueous electrolyte is low due to the surface oxidation. Hence, a passivation layer on Si is required to cope against the instability at the oxidative potential and poor photoelectrochemical performance of Si [10]. Thus, the introduction of a co-catalyst to increase the PEC performance and chemical stability is desirable.

Generally, for the direct deposition of GO onto diverse substrates, high-cost vacuum plasma has been utilized instead of the atmospheric plasma in the deposition technique. In our previous work, to deposit the N-doped monolayer graphene on the Si layer photocathode, copper substrate was used to grow the N-doped graphene during chemical vapor deposition [13,14]. Herein, we propose a facile, easy, and novel synthesis strategy using atmospheric plasma to deposit rGO onto the p-type Si (p-Si) directly without any copper substrate. This technique can also be used for the effective deposition of rGO on various other substrates. The deposited rGO shows hydrophobic surface properties. A mixture of methane and Ar gas was streamed to stimulate the formation of hydrophobic rGO on the wafer, while the preparation of hydrophilic GO was inspired under additional Ar plasma treatment. The method of direct deposition of catalysts onto the substrate can reduce interfacial issues between the photoelectrode and co-catalyst, and minimize photo corrosion and photooxidation due to the defects obtained through plasma action. In this study, we suggest a new pathway for the direct deposition of GO and rGO onto Si substrates. The prepared hydrophobic rGO and hydrophilic GO are applied as PEC co-catalysts for hydrogen production to confirm the correlation between the surface properties and catalytic performance.

## 2. Materials and Methods

### 2.1. Synthesis of Graphene Oxide

Silicon (Si) wafer substrate of 4 mm × 4 mm (1 Ω) was prepared for the deposition of reduced graphene oxide (rGO) and graphene oxide (GO). The surface was ultrasonically cleaned in ethanol for 10 min to remove the adherent contaminants. Both rGO and GO were deposited using a high-density atmospheric plasma generator PGS-300 (Expantech Co., Suwon, Korea) operated at 900 MHz. The plasma was generated with the power of 240 W (900 MHz) using argon gas at a rate of 4 l/m as a plasma carrier. A mixture of methane (10%) and argon gas was introduced to the plasma at a rate of 60 SCCM. Followed by the automatic dissociation of methane into carbon, rGO was deposited on the p-Si wafer for six minutes. The GO was obtained by the additional treatment of plasma with argon gas as the carbon source for one min. To eradicate the back contact of the photoelectrode, epoxy was spread on the sample.

### 2.2. Characterization of Materials

High-Performance X-ray Photoelectron Spectroscopy (XPS, Model: K-ALPHA +, Source aluminum (Al)) with Kα X-ray analyzer was used to study the various functional groups of hydrophobic rGO and hydrophilic GO deposited on titanium substrate. The morphology of the carbon-coated samples was investigated by Field Emission Transmission Electron Microscopy (FE-TEM, Model: JEM-2100F, JEOL LTD). Wettability was measured by the contact angle machine (Phoenix 300, SEO Korea) using distilled water in stationary mode. Laser Raman spectroscopy (NRS-5100) was used with the laser wavelength of 532 nm to distinguish between graphene and GO and find the number of layers.

### 2.3. Photoelectrochemical Measurements

The assessment of the photoelectrochemical properties was performed by the bi Potentiostat (CHI 760E, CH Instruments, Inc. Austin, TX, USA). In a three-electrode configuration, Pt wire was used as the counter electrode, and an Ag/AgCl/3 M NaCl electrode was used as the reference electrode. The reference electrode was carefully attuned concerning the RHE at 25 °C in an aqueous 1 M perchloric acid solution, which was saturated with pure H_2_ gas. The RHE was calibrated between −0.201 V and −0.203 V vs. the Ag/AgCl reference electrode. The 300 W Xe lamp with a light intensity of 100 mW cm^−2^ using a glass Air Mass 1.5 Global filter was used as the source of visible light for radiance on the substrate.

## 3. Results and Discussion

Chemical composition analysis was carried out by the Raman spectroscopic and X-ray photoelectron spectroscopic technique. From the Raman spectroscopy results, it was confirmed that the as-deposited rGO and GO were distinguishable from graphene. The typical peaks for graphene are D (1350 cm^−1^), G (1580 cm^−1^), and a 2D peak at 2690 cm^−1^. Pristine graphene does not show any D peak, which represents edges of a graphene crystal and chemical bonds [15], while typical Raman spectra for GO are characterized by its D and G band corresponding to 1353 cm^−1^ and 1605 cm^−1^, respectively [16]. Raman spectrum in Figure 1a shows D, G, and 2D peaks at 1345 cm^−1^, 1585 cm^−1^, and 2685 cm^−1^, respectively, which indicates that rGO was successfully deposited. There is a slight redshift in G peak due to multi-layer deposition stacked one above the other and by defects such as vacancies and grain boundaries [17,18,19,20]. The intensities of D, G, and 2D peaks for Figure 1a are 351, 311, and 268, respectively, and the intensity ratio of I2D/IG is 0.86, which shows multi-layered reduced graphene oxide [21]. Raman spectrum in Figure 1b shows D, G, and 2D peaks at 1352 cm^−1^, 1593 cm^−1^, and 2689 cm^−1^, respectively. The shifted peaks compared to the peaks in Figure 1a matched closely to the peak in hydrophilic GO [16]. The intensities of D, G, and 2D peaks for Figure 1b are 460, 358, and 300, respectively, and the ratio of I2D/IG is 0.83, which indicates multi-layered deposition of graphene oxide [21]. Similarly, the intensity ratio of I(D)/I(G) for rGO in Figure 1a is 1.12 and the intensity ratio of I(D)/I (G) for GO is 1.28, which is used as an index for structural disorder. The increment in I(D)/I (G) ratio for GO is an indication of substantial functionalization of GO, confirmed by X-ray Photoelectron Spectroscopy [22].

From the XPS results, prepared hydrophilic GO and hydrophobic rGO was confirmed to combine more various functional groups. The XPS spectra survey scan shows the deposited hydrophobic rGO layer has carbon and oxygen elements as in Figure 2a. The high-resolution XPS spectra were acquired for C1s peak and O1s peak as in Figure 2b,c, respectively. The C1s peak in Figure 2b is resolved with two peaks, denoting C–C bonding located at 284.8 eV and C–O bonding at 285.4 eV [23,24]. Figure 2c shows the O1s peak split into two main peaks. The primary component is corresponding to C–OH bonding and C–O–C bonding at 532.8 eV and the secondary component represents a carbon double bond with oxygen, i.e., C=O at 531 eV. The atomic percentage ratio of C/O calculated from XPS survey spectra in Figure 2 is 6.57, which corresponds to rGO [25,26,27].

The hydrophobic rGO was treated with argon plasma for one minute after carbon deposition and resulted in the hydrophilic GO. A notable variation in the XPS spectra of the hydrophilic GO is witnessed with the addition of oxygen-containing functional groups. When comparing the XPS spectra survey scan in Figure 2a with Figure 3a, a noticeable increment occurs in O1s peak. The high-resolution XPS spectra of C1s peak in Figure 3b shows various oxygen-containing groups compared to hydrophobic rGO due to diverse precursors of plasma treatment. Different functional groups revealed in C1s peak are at 284.75 eV (C–C), 285.73 eV (C–OH), 286.35 eV (C=O and C–OH), 287.44 eV (C=O), and 289.07 eV containing carboxylates (O–C=O), carbonyl (–C=O), and carboxylic acid (O=C–OH). Figure 3c shows O1s peak containing two main peaks at 530.77 eV (C=O) and 532.73 eV (C–OH and C–O–C), and Figure 3d shows N1s peak at 400.6 eV and 402.4 eV are well-matched with XPS database corresponding to C–N–H and C–N, respectively. The proportion of oxygen in hydrophilic GO treated with argon plasma is increased from 13.21% (untreated) to 27.27% (treated) with the addition of N1s peak. The atomic percentage ratio of C/O calculated from XPS survey spectra in Figure 3 is 2.63 corresponding to GO [25,26,27].

This argon plasma treatment resulting in the formation of oxygen-containing functional groups makes the carbon deposited material more hydrophilic by adding oxygen functional groups onto the surface of GO deposited on the silicon (Si) electrode. The reason behind the transition from hydrophobic to hydrophilic is that energy from argon plasma is transferred to the deposited rGO by radiation (a distance of 1 cm was maintained between the plasma flame and substrate). As a result of the energy transfer, sp^2^ carbon atoms broke, allowing oxygen atoms from the environment to react on the surface of carbon atoms making it hydrophilic. A minute amount of nitrogen atoms were also added from the environment as our process was operated in an open environment. The transition of the rGO surfaces from hydrophobic to hydrophilic using this easy and straightforward argon plasma treatment was conducted in a cost-optimized method without applying any temperature and vacuum formation. The mechanism that made the surface hydrophilic is the ion energy in plasma, which is usually higher than 10 eV exceeding the binding energy for carbon: 2.7 eV and 3.6 eV for π-bonding and σ-bonding, respectively, forming oxygen-containing functional groups [17]. Functional groups like C–OH, C–H, C–O–C, –COOH, C–O–C, C–OH, –C=O, C=O, and carboxylates (O–C=O) make hydrophilic GO surfaces due to dynamic interaction with water by generating intermolecular forces called hydrogen bonding [28,29,30,31]. Figure 4 shows TEM images of obtained rGO in order to investigate its morphology, and the images were examined through FE-TEM to confirm that rGO was formed well. In Figure 4a, the representing FE-TEM images showing that transparent multi-layers are stacked one above the other with no crumples and ripples are because the layers are smoothly deposited by plasma [32]. The appearance of these crumples usually appears in the exfoliation and restacking processes that create the deformation of the stacked layers [33]. Figure 4b shows that rGO contains multi-layers and the inset is the selected area diffraction showing polycrystalline structure, corresponding to the (002) rGO plane.

The wettability of hydrophobic rGO and hydrophilic GO was measured through a contact angle machine using D.I water droplet in a stationary mode. The average and Standard deviation (STD ) of water contact angle were taken for three samples on the surface of superhydrophobic, hydrophobic, and hydrophilic are given (accurate values are described in Table 1).

Hydrophobic rGO was coated without the addition of argon plasma treatment. The hydrophobic rGO surface was treated with argon plasma for one minute and the contact angle reduction from hydrophobic rGO to hydrophilic GO was due to the addition of oxygen-containing functional groups causing hydrogen bonding with water molecules.

To confirm the applicability of this synthesis procedure, the GO deposited on p-type Si (p-Si) was utilized in photoelectrochemical (PEC) for the hydrogen evolution reaction (HER). Firstly, the current density was measured as the electric current flowing per-unit cross-sectional area of a material in 1 M of aqueous perchloric acid (HClO_4_) (pH = 0) as the potential stroked from 0 V to −0.6 V vs. RHE in a three-electrode cell. The J-E curves from Figure 5a shows the hydrogen evolution capability of our samples under the dark condition toward PEC hydrogen production. As the negative (cathodic) potential is applied, the magnitude of the current density increases, implying that the negative potential twitched the reduction reaction at the cathode. The onset potentials, which represent the starting point of the current increment (−0.1 mA cm^−2^), were observed for all the three samples. The observed onset potential values, which indicates current density of −0.1 mA cm^−2^ under dark condition are as follows: a bare p-Si electrode is −0.85 V, hydrophilic GO electrode is −0.66 V, and hydrophobic rGO electrode is −0.62 V, respectively. Based on the HER onset potential values, hydrophobic rGO started progressing earlier than hydrophilic GO and bare p-Si. Followed by the onset potential, the current density increases gradually with the negative applied potential. The current density of the p-Si wafer is −0.4 mA cm^−2^. It was further increased by the coated rGO on p-Si wafer, hydrophobic rGO of −0.15 V, and hydrophilic GO of −0.25 V, respectively. This implies that the p-Si wafer showed better properties with rGO as co-catalyst since it forms a passivating layer that blocks photo corrosion and photooxidation. Hydrogen production under the illumination of light was found with the inspection of linear sweep voltammetry (LSV). In our measurement setup, a 300 W Xe lamp was used with a light intensity of 100 mW cm^−2^ and the light passed through an Air Mass 1.5 global filter in 1 M of aqueous perchloric acid (HClO_4_) (pH = 0). Figure 5b shows the photoelectrochemical data of our samples under the light illumination. Generally, on the subject of visible light introduction on the catalyst surface during the LSV, an augmentation of current density has been observed because of the photo-generated electron by absorption of light in the visible region [34]; however, due to the issues that exist at the interface of the p-Si wafer and the electrolyte, the onset potential of the p-Si wafer was raised to −0.6 V vs. RHE. Differing from the current density loss in p-Si wafer, the current density of the hydrophilic GO (−0.33 V vs. RHE) and hydrophobic rGO (−0.1 V vs. RHE) was improved. From this outcome, it is clear that the incorporated rGO on p-Si wafer as an effective layer would enhance the activity by inhibiting the silicon oxide formation and corrosion. Thus, Ar plasma treatment promotes the chemical reaction between the solution and the surface and increases the HER charge transfer in the PEC. The hydrophobic rGO deposited on p-Si wafer shows the limiting current density of −3.43 mA cm^−2^. The current increment between dark and light condition at −0.2 V to −0.4 V is calculated and bare p-Si possess 0.004 V, which is less than that of hydrophobic rGO (2.493 V) and hydrophilic GO (0.225 V). Figure 5c approved the increment of current in rGO bearing p-Si wafer due to the higher surface activation of p-Si wafer. As shown in Figure 5d, the photovoltage is a necessary factor to estimate the number of onset potentials increased by finding the difference between the onset potentials under the dark and illuminated condition. At −0.1mA cm^−2^, there is betterment in photovoltage of hydrophobic rGO (0.52 V) in relation with the hydrophilic GO (0.3 V of photovoltage) and the bare p-Si electrode (0.16 V of photovoltage) (accurate values are described in Table 2). The extrapolated value is calculated through a linear sweep potential from −0.5 V to −0.6 V, and saturation current density of −3.5 mA cm^−2^ shown in Figure 5b. On top of this, the onset potential reduction gradation from bare p-Si to hydrophobic rGO in both light and dark conditions is approximately 2.5 times higher.

Impedance measurements were performed to study the resistance prevails in the charge transfer of hydrophobic rGO and hydrophilic GO on the p-Si wafer with a frequency of 10^6^−1 Hz, an amplitude of 5 mV, and at 0 V vs. RHE in a three-electrode system. The hydrophobic rGO and hydrophilic GO show arcs that are smaller than the arc exhibited by the bare p-Si electrode as in Figure 6a. Usually, the high-frequency region embodies the series resistance (R_s_), which is the electrolyte and interfacial resistance, and low-frequency county indicates the charge transfer resistance (R_ct_), which is the transport resistance between photocathode and electrolyte [36]. Figure 6b signifies the Nyquist plot representing a typical impedance result. Corresponding to the LSV figures, the smaller arcs in the Nyquist plots of both GO materials conveyed less resistance between GO deposited on p-Si wafer and electrolyte. The R_ct_ values obtained by fitting with an equivalent circuit are tabulated (accurate values are described in Table 3). The charge carrier transfer has taken place at a higher rate between the photo absorber and electrolytic solution with the reduction of kinetic barriers across the double layer. Consequently, GO might promote faradaic reactions by reducing the charge transfer resistance in the semiconductor depletion layer, which leads to a higher photocurrent response because of the higher band bending in the depletion layer.

Further, capacitance measurements of the hydrophobic rGO and hydrophilic GO deposited on Si electrodes were performed as the potential was swept from 0.6 V to −0.5 V vs. RHE in a three-electrode cell without illumination. Build on the capacitance results, the flat band potential (E_fb_) of the hydrophobic rGO and hydrophilic GO deposited on Si electrodes were calculated using the Mott–Schottky relationship:1/C_SC_^2^ = 2(E–E_fb_-kT/e)/(eεε_0_N),(1)
where C_SC_ is the capacitance of the space charge region, ε is the dielectric constant of the semiconductor, ε_0_ is the permittivity of free space, N is the donor density (electron donor concentration for an n-type semiconductor or hole acceptor concentration for a p-type semiconductor), E is the applied potential, and E_fb_ is the flat band potential. E_fb_ was determined by extrapolating the capacitance information. From the slope value, donor density can be determined. According to this relation, hydrophilic GO on p-Si exhibits flat band potential of 0.11 V vs. RHE, whereas hydrophobic rGO on p-Si shows 0.25 V vs. RHE. Higher band bending at the interface between electrode and electrolyte speeds up the electron transfer on to the surface, employing faster charge separation of generated electrons and holes [37]. The possibility of charge recombination or surface trapping at sub-bandgap energy levels may also be diminished.

Overall, it is evident that rGO as the co-catalyst of the p-Si photocathode is capable of encouraging the reaction kinetics between the interfacial layers of the buried electrode and electrolyte and acts as the passivation layer that preserves the p-Si photocathode from corrosion and native oxide formation. It is obvious from the gained results that with rGO deposited on p-Si, the minority carriers generated by the optical absorption can move with no trouble and participate in the evolution of hydrogen more rapidly than in the bare p-Si electrode. Therefore, rGO optimizes the negative effects of the interfacial layer between the p-Si and electrolyte in an integrated fashion to harvest more hydrogen, making it an excellent baseline co-catalyst to investigate in PEC cell.

## 4. Conclusions

In summary, we have reported a simplistic method to formulate the reduced graphene oxide (rGO) decorated p-type Si electrode for the efficient photoelectrochemical (PEC) hydrogen evolution reaction (HER) under atmospheric plasma. The prepared electrodes were keenly scrutinized through the basic photoelectrochemical characterizations. Besides, both the electrode of hydrophobic rGO and hydrophilic GO deposited on the Si electrode showed the enhanced production of hydrogen relative to the bare p-Si electrode. Especially, hydrophobic rGO deposited on p-Si electrode emboldens a better activity toward HER in PEC cell.

## Figures and Tables

**Figure 1 materials-13-00012-f001:**
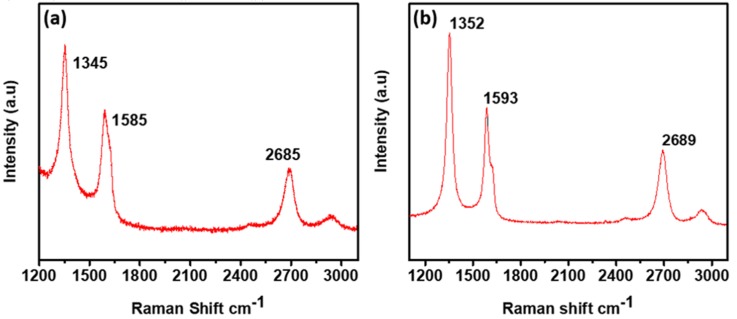
(**a**) Raman spectrum of as-deposited reduced graphene oxide and (**b**) the spectrum of the graphene oxide produced by the additional argon plasma treatment.

**Figure 2 materials-13-00012-f002:**
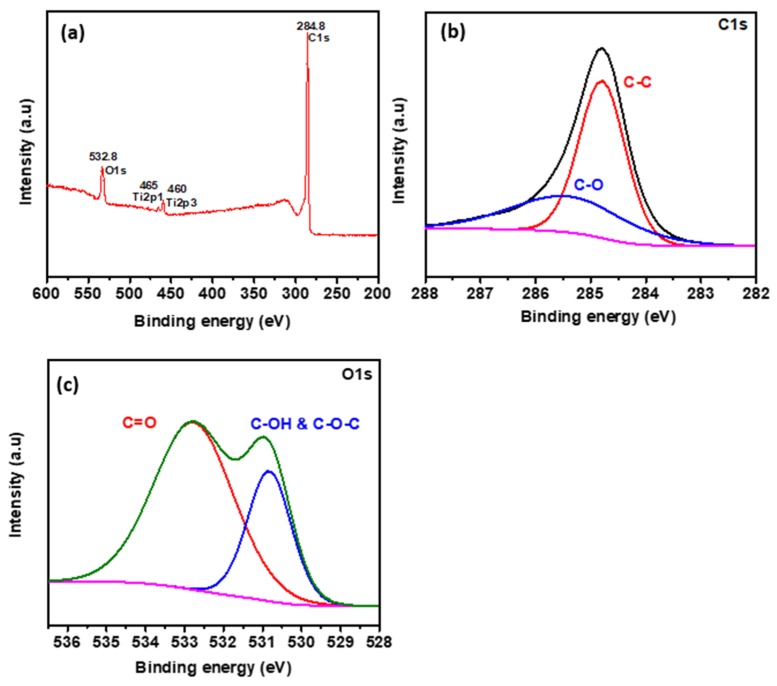
The XPS spectra survey of the as-deposited hydrophobic reduced graphene oxide (rGO). (**a**) XPS wide-scan spectrum and (**b**) XPS high-resolution spectrum registered for the energy range of C1s and (**c**) O1s.

**Figure 3 materials-13-00012-f003:**
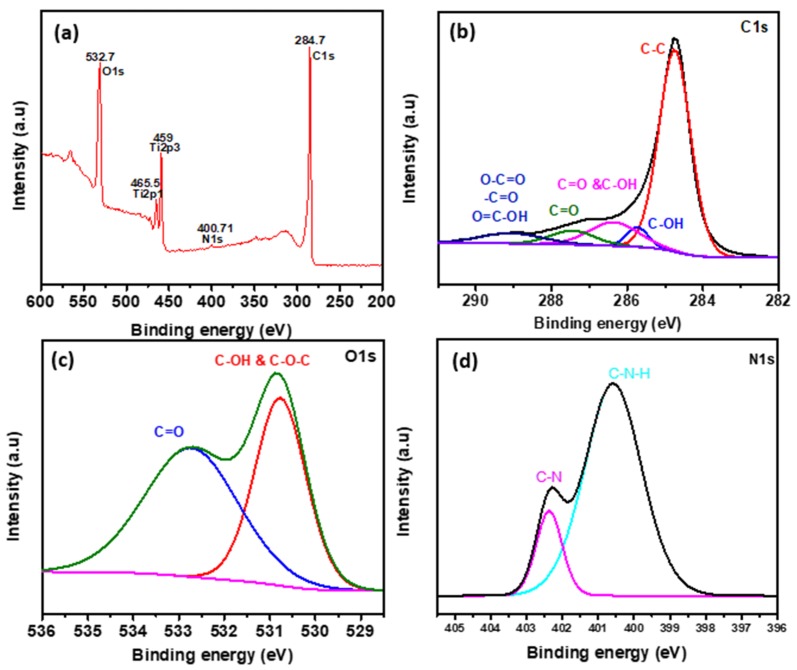
The XPS spectra survey of the hydrophilic graphene oxide (GO). (**a**) XPS wide-scan spectrum, (**b**) XPS high-resolution spectrum registered for the energy range of C1s, (**c**) O1s, and (**d**) N1s spectra of GO.

**Figure 4 materials-13-00012-f004:**
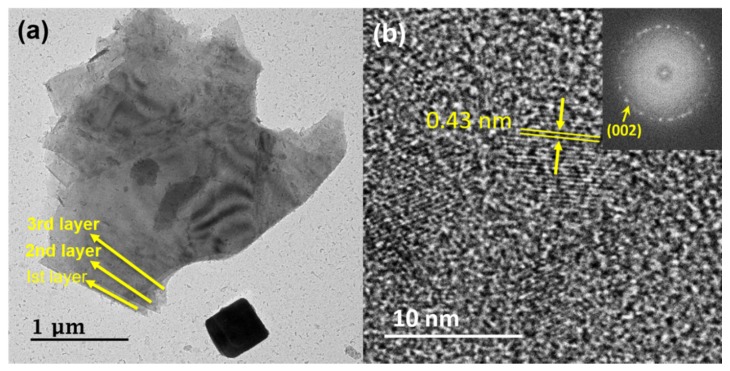
Field Emission Transmission Electron Microscope images of GO by atmospheric plasma. (**a**) Showing transparent sheets on top of one another. (**b**) Showing multi-layers and selective area electron diffraction image (inset).

**Figure 5 materials-13-00012-f005:**
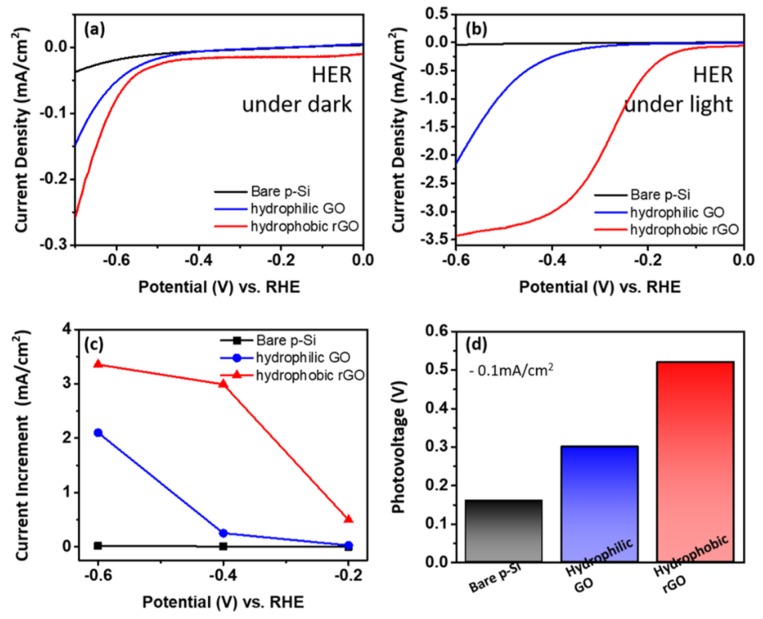
Comparison of the electrochemical activity of hydrophilic GO and hydrophobic rGO deposited on silicon (Si) electrodes. (**a**,**b**) Photocurrent density—potential curves of bare p-Si, hydrophilic GO, and hydrophobic rGO under dark and light conditions and the potential swept from 0 V to –0.6 V vs. RHE in a three-electrode cell. (**c**) Current increment between dark and light conditions. (**d**) Photovoltage of bare p-Si, hydrophilic GO, and hydrophobic rGO at −0.1 mA cm^−2^.

**Figure 6 materials-13-00012-f006:**
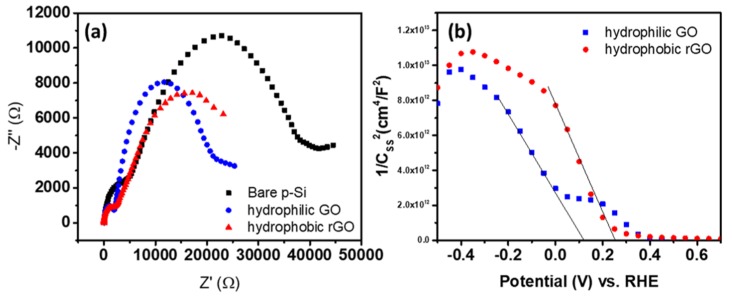
Comparison of the electrochemical activity of hydrophilic GO and hydrophobic rGO deposited on silicon (Si) electrodes. (**a**) Mott–Schottky plots from capacitance measurement as a function of potential vs. RHE under dark conditions. (**b**) Nyquist plot for bare Si, hydrophilic GO, and hydrophobic rGO on Si electrode at 0 V vs. RHE.

**Table 1 materials-13-00012-t001:** Average and STD of water contact angle taken for three samples on the surface of superhydrophobic, hydrophobic, and hydrophilic.

Surface	Average (°)	STD
**Hydrophobic rGO**	67	12.5
**Hydrophilic GO**	14.3	2.3
**Si wafer without coating**	53.2	8.3

**Table 2 materials-13-00012-t002:** Summary of the experimental data of bare p-type Si (p-Si), hydrophilic GO, and hydrophobic rGO as co-catalyst on Si electrode.

Condition		Bare p-Si	Hydrophilic GO	Hydrophobic rGO
**Onset potential at dark condition (V)**		−0.85 ^†^	−0.66	−0.62
**Potential at −0.1 mA/cm^2^ at light condition (V)**		−0.69 ^†^	−0.33	−0.1
**Photovoltage* (V)**		0.16	0.3	0.52
**Current increment at specific potential between dark and light condition (mA/cm^2^)**	at −0.2 V_RHE_	0.003	0.026	0.498
	at −0.4 V_RHE_	0.007	0.251	2.991
	at −0.6 V_RHE_	0.018	2.1	3.353

* The photovoltage is defined as the difference between the onset potential under the dark and illumination condition referring to the literature [35]. ^†^ Values were measured and extrapolated by our group referring to the figures.

**Table 3 materials-13-00012-t003:** Obtained from the fitting of the electrochemical impedance spectroscopy (EIS) data.

Sample	R_overall_ (Ω cm^2^)	R_2_ (Ω cm^2^)	C_2_ (F/cm^2^)	R_1_ (Ω cm^2^)	C_1_ (F/cm^2^)	Error on the Fitting Data
Bare p-Si	23.05	7169	6.221 × 10^−8^	-	-	0.2212
Hydrophilic GO	1.848	2108	6.762 × 10^−7^	1.809 × 10^4^	3.165 × 10^−5^	0.3561
Hydrophobic rGO	0.4963	2561	1.809 × 10^−6^	1.382 × 10^4^	1.229 × 10^−4^	0.105

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
