# Peer review of "In-situ Deposition of Graphene Oxide Catalyst for Efficient Photoelectrochemical Hydrogen Evolution Reaction Using Atmospheric Plasma"

_materials, 2019, doi:10.3390/ma13010012_

Round 1

Reviewer 1 Report

Attached

Author Response

Title: In-situ Deposition of Graphene Oxide Catalyst for Efficient Photoelectrochemical Hydrogen Evolution Reaction using Atmospheric Plasma

Authors: Khurshed Alam, Yelyn Sim, Ji-Hun Yu, Janani Gnanaprakasam, Hyeonuk Choi, Yujin Chae, Uk Sim*, and Hoonsung Cho*

We appreciate the editors and referees for providing valuable comments and suggestions to improve the quality of our recent submission for the consideration of publication by Applied Surface Science. According to the reviewers’ comments, we have thoroughly revised this manuscript. The corrections are highlighted in red in the main manuscript. The followings are our point-by-point responses:

Thank you for your kind consideration.

Response to Reviewers’ Comments

Reviewer: 1

The authors report on the fabrication of hydrophobic and hydrophilic graphene oxide and its applications on PEC co-catalyst hydrogen production. The results have presented systematically. The characteristic properties of these GO and its effect on PECs are interesting. I recommend this manuscript for publication after addressing the following comment.

Please mention the Y-axis in Fig. 2 and 3.

Response:
We apologize for those minor mistakes, sincerely. In response to your comment, we insert the information of Y-axis in Fig.2 and 3. (page 4)

2. The XPS spectra survey of the as-deposited hydrophobic rGO on Si substrate. XPS wide-scan spectrum (a) and XPS high-resolution spectrum registered for the energy range of C1s (b) and O1s (c).

(page 5)

Figure. 3. The XPS spectra survey of the hydrophilic GO on Si substrate. XPS wide-scan spectrum (a), XPS high-resolution spectrum registered for the energy range of C1s (b), O1s (c), and N1s spectra of GO (d). The authors have mentioned the hydrophilicity and hydrophobicity of fabrication graphene oxide. Does the plasma treatment time affect the hydro-phobic/philic characteristics?

Response:
In response to your precious comment, the hydrophobic and hydrophilic characters are independent of time. We have observed that transition from hydrophobic to hydrophilic characters can only be obtained by argon plasma treatment alone while turning carbon source gas off. The reason behind transition from hydrophobic to hydrophilic is that energy from argon plasma is transferred to deposited rGO by radiation keeping distance of 1 cm, as a result transferring energy is breaking sp2 carbon atoms, allowing oxygen atoms from the environment to react on the surface of carbon atoms that makes it hydrophilic as well as minute amount of nitrogen atoms are also added from the environment as our process is operated in an open environment.

We mention that information in our manuscript. (page 5, line No:161-166)

The reason behind transition from hydrophobic to hydrophilic is that energy from argon plasma is transferred to deposited rGO by radiation keeping distance of 1 cm, as a result transferring energy is breaking sp2 carbon atoms, allowing oxygen atoms from the environment to react on the surface of carbon atoms that makes it hydrophilic as well as minute amount of nitrogen atoms are also added from the environment as our process is operated in an open environment. Please mention the error on the fitting data in Table 3.

Response:
Thank you for your kind comments. we have inserted an error value in Table 3 that shows the difference between the EIS measurement data and the fitting data. (page 9, line No:255)

Table 3. Obtained from the fitting of the EIS data.

Sample

Roverall
(Ωcm2)

R2
(Ωcm2)

C2
(F/cm2)

R1
(Ωcm2)

C1
(F/cm2)

Error on the fitting data

Bare p-Si

23.05

7169

6.221E-08

0.2212

Hydrophilic GO

1.848

2108

6.762E-07

1.809E04

3.165E-05

0.3561

Hydrophobic rGO

0.4963

2561

1.809E-06

1.382E4

1.229E-04

0.105

Please, revise the writing. It is very difficult to get the meaning in many places.

Response:
Thank you for pointing out these crucial mistakes. In response to your comment, we have revised our manuscript to correct ambiguity.

(page 7-8, line No:215-226)

On the basis of the HER onset potential values, hydrophobic rGO started progressing earlier than hydrophilic GO, and bare p-Si. Followed by the onset potential, the current density increases gradually with the negative applied potential. The current density of p-Si wafer is -0.4 mA cm-2. It was further increased by the coated rGO on p-Si wafer, hydrophobic rGO of -0.15 V and hydrophilic GO of -0.25 V respectively. It hints that the problematic p-Si wafer behaved healthier with rGO as the co-catalyst and also the passivating layer by condensing the photo-oxidation and photo-corrosion.

(page 8, line No:234-238)

Differing from the current density loss in p-Si wafer, the current density of the hydrophilic GO (-0.33 V vs. RHE) and hydrophobic rGO (-0.1 V vs. RHE) is improved. From this outcome, it is clear that the incorporated rGO on p-Si wafer as an effective layer would enhance the activity and inhibit the silicon oxide formation and corrosion.

(page 10, line No:290-297)

 Overall, it is evident that rGO as the co-catalyst of the p-Si photocathode is capable of encouraging the reaction kinetics between the interfacial layers of the buried electrode and electrolyte and acts as the passivation layer that preserves the p-Si photocathode from corrosion and native oxide formation. It is obvious from the gained results, in rGO deposited p-Si, the minority carriers generated by the optical absorption can move with no trouble and participate in the evolution of hydrogen more rapidly than in the bare p-Si electrode. Therefore, rGO optimizes the negative effects of the interfacial layer between the p-Si and electrolyte in an integrated fashion to harvest more hydrogen, making it an excellent baseline co-catalyst to investigate in PEC cell.

Reviewer 2 Report

This article deals with the formation of reduced graphene oxide layer on p-Si employing atmospheric Ar plasma. The subject of this work is appropriate for publication in Materials. However, several revisions described below should be carried out prior to acceptance for publication. 

(1) XPS results: In the case of pydrophilic GO, nitrogen (C-N-H and C-N) was detected in addition to carbon and oxygen. What is the source of nitrogen? (N2 in air?) Chemical process of the nitrogen introduction is unclear because the plasma has been generated using CH4 and Ar. Moreover,  unknown peak appears aorund 470 eV for both rGO and GO. Assignment of the peaks sould be shown.

(2) Photocurrent density - porential curve: the onset potential of p-Si has been reported to be -0.85 V vs. RHE. How was the potential value found? The potential has been scanned between 0 and -0.6 V (Figure 5). 

(3) Raman spectra: the intensity ratios of I(2D)/I(G) have been given to show the multi-layered deposition of rGO and GO. In Figure 1, D-band was also observed. The intensity ratio I(D)/I(G) has been widely accepted as a index of structural disorder of carbon materials. To specify the sutuctural disorder, the I(D)/I(G) should be given and discussed because the recorded D-bands are quite significant.

(4) Methods: the X-ray used for XPS experiments was described as "Kα". In this case, what is the X-ray source? (Mg or Al?)

(5) Spelling, description of parenthesis and punctuations, proper use of superscript/subscript, and usage of capital/lowercase letters (e.g. journal names) shoud be carefully re-checked and corrected. 

Author Response

Title: In-situ Deposition of Graphene Oxide Catalyst for Efficient Photoelectrochemical Hydrogen Evolution Reaction using Atmospheric Plasma

Authors: Khurshed Alam, Yelyn Sim, Ji-Hun Yu, Janani Gnanaprakasam, Hyeonuk Choi, Yujin Chae, Uk Sim*, and Hoonsung Cho*

We appreciate the editors and referees for providing the valuable comments and suggestions to improve the quality of our recent submission for the consideration of publication by Applied Surface Science. According to the reviewers’ comments, we have thoroughly revised this manuscript. The corrections are highlighted in red in the main manuscript. The followings are our point-by-point responses:

Thank you for your kind consideration.

Response to Reviewers’ Comments

Reviewer: 2

This article deals with the formation of reduced graphene oxide layer on p-Si employing atmospheric Ar plasma. The subject of this work is appropriate for publication in Materials. However, several revisions described below should be carried out prior to acceptance for publication.

(1) XPS results: In the case of pydrophilic GO, nitrogen (C-N-H and C-N) was detected in addition to carbon and oxygen. What is the source of nitrogen? (N2 in air?) Chemical process of the nitrogen introduction is unclear because the plasma has been generated using CH4 and Ar. Moreover, unknown peak appears aorund 470 eV for both rGO and GO. Assignment of the peaks sould be shown.

 Response:

We apologize for those minor mistakes. We did not use any source for nitrogen, nitrogen atoms are added from the environment as our process is operated in an open environment. We have performed XPS analysis using titanium substrate as a substrate.

(2) Photocurrent density - potential curve: the onset potential of p-Si has been reported to be -0.85 V vs. RHE. How was the potential value found? The potential has been scanned between 0 and -0.6 V (Figure 5).

 Response:
        Thank you for your kind comments. The values were measured and extrapolated by our group referring to the figure 5. And as response to your comments, we insert above information in the manuscript

(page 8, line No:221)

Table 2. Summary of the experimental data of bare p-Si, hydrophilic GO and hydrophobic rGO as co-catalyst on Si electrode.

Condition

Bare p-Si

Hydrophilic
GO

Hydrophobic
rGO

Onset potential at dark condition (V)

-0.85

-0.66

-0.62

Potential at -0.1mA/cm2 at light condition (V)

-0.69

-0.33

-0.1

Photovoltage* (V)

0.16

0.3

0.52

Current increment at specific potential between dark and light condition (mA/cm2)

at 0.2VRHE

0.003

0.026

0.498

at 0.4VRHE

0.007

0.251

2.991

at 0.6VRHE

0.018

2.1

3.353

* : The photovoltage is defined as the difference between the onset potential under the dark and illumination condition referring to the literature [34].
: Values were measured and extrapolated by our group referring to the figures

(3) Raman spectra: the intensity ratios of I(2D)/I(G) have been given to show the multi-layered deposition of rGO and GO. In Figure 1, D-band was also observed. The intensity ratio I(D)/I(G) has been widely accepted as a index of structural disorder of carbon materials. To specify the sutuctural disorder, the I(D)/I(G) should be given and discussed because the recorded D-bands are quite significant.

 Response:

Thank you for your kind comments. I am agreed with your statement that D peak is an indication of structural disorder so I am adding ratio of I(D)/I(G) in both rGO and GO. As intensities ratio of I(2D)/I(G) is calculated and given showing multi-layered deposition. The intensity ratio of I(D)/I(G) for rGO in Fig. 1 (a) is 1.12. Similarly, intensity ratio of I(D)/I (G) for GO is 1.28 that is used as an index for structural disorder [16]. This significant D peak is expected to occur because our process for the synthesis of rGO and GO is atmospheric based and without using any catalyst to grow both rGO and GO. The increment in I(D)/I (G) ratio for GO is an indication of substantial functionalization of GO, confirmed by X-ray photoelectron spectroscopy [22]

[16] Fatima Tuz JohraJee-Wook LeeWoo-Gwang, J., Facile and safe graphene preparation on solution ba sed platform. Journal of Industrial & Engineering Chemistry 2014, 20, (5), 2883.

[22] Tran, N. A.; Lee, C.; Lee, D. H.; Cho, K.-H.; Joo, S.-W., Water Molecules on the Epoxide Groups of Graphene Oxide Surfaces : Water Molecules on the Epoxide Groups. Bulletin of the Korean Chemical Society 2018, 39, (11), 1320-1323.

And we insert those information in our manuscript

(page 3-4, line No:126-132)

Similarly, the intensity ratio of I(D)/I(G) for rGO in Fig. 1 (a) is 1.12 and the intensity ratio of I(D)/I (G) for GO is 1.28 that is used as an index for structural disorder.The increment in I(D)/I (G) ratio for GO is an indication of substantial functionalization of GO, confirmed by X-ray photoelectron spectroscopy.[22]

(4) Methods: the X-ray used for XPS experiments was described as "Kα". In this case, what is the X-ray source? (Mg or Al?)

 Response:

Thank you for your kind comments. The X-ray source we used for analysis is aluminum (Al).

(5) Spelling, description of parenthesis and punctuations, proper use of superscript/subscript, and usage of capital/lowercase letters (e.g. journal names) should be carefully re-checked and corrected.

Response:
        Thank you for pointing out these crucial mistakes. In response to your comment, we have revised our manuscript to correct ambiguity.

(page 1, line No:43)

Photoelectrochemical cell (PEC) which can produce H2 production by solar energy-driven photocatalysis is the effectual device for the production of hydrogen.

(page 3, line No:111-123)
Chemical composition analysis was carried out by the Raman spectroscopic and X-ray photoelectron spectroscopic technique. From the Raman spectroscopy results, it was confirmed that the as-deposited rGO and GO were distinguishable from graphene. The typical peaks for graphene are D (1350 cm-1), G (1580 cm-1) and 2D peak at 2690 cm-1. Pristine graphene does not show any D peak which represents edges of a graphene crystal and chemical bonds [12], while typical Raman spectra for GO are characterized by its D and G band corresponding to 1353 cm-1 and 1605 cm-1 respectively [13]. Raman spectrum in Fig. 1 (a) shows D, G and 2D peaks at 1345 cm-1, 1585 cm-1 and 2685 cm-1 respectively which indicates that rGO was successfully deposited. A slight redshift in G peak due to multi-layer deposition upon one another and by defects such as vacancies, grain boundaries [14-17]. The intensities of D, G and 2D peaks for Fig. 1 (a) are 351, 311 and 268 respectively and the intensity ratio of I2D/IG is 0.86 which shows multi-layered reduced graphene oxide [18]. Raman spectrum in Fig. 1 (b) shows D, G and 2D peaks at 1352 cm-1, and 1593 cm-1 and 2689 cm-1 respectively. The shifted peaks compared to the peaks in Fig. 1 (a) matched closely to the peak in hydrophilic GO [13]. The intensities of D, G and 2D peaks for Fig. 1 (b) are 460, 358, and 300 respectively and the ratio of I2D/IG is 0.83 indicated multi-layered deposition of graphene oxide [18]. The intensity of D peak in hydrophilic GO is attributed to substantial functionalization of GO, which is confirmed by XPS analysis [19].

(page 5, line No:151)

Functional groups like C-OH, C-H, C-O-C, -COOH, C-O-C, C-OH, –C=O, C=O, and carboxylates (O-C=O) make hydrophilic GO surface due to dynamic interaction with water by generating intermolecular forces called hydrogen bonding

(page 7-8, line No:215-226)
On the basis of the HER onset potential values hydrophobic rGO started progressing earlier than hydrophilic GO, and bare p-Si. Followed by the onset potential, the current density increases gradually with the negative applied potential. The current density of p-Si wafer is -0.4 mA cm-2. It was further increased by the coated rGO on p-Si wafer, hydrophobic rGO of -0.15 V and hydrophilic GO of -0.25 V respectively. It hints that the problematic p-Si wafer behaved healthier with rGO as the co-catalyst and also passivating layer by condensing the photo-oxidation and photo-corrosion.

(page 8, line No:234-238)
Differing from the current density loss in p-Si wafer, the current density of the hydrophilic GO (-0.33 V vs. RHE) and hydrophobic rGO (-0.1 V vs. RHE) is improved. From this outcome, it is clear that the incorporated rGO on p-Si wafer as an effective layer would enhance the activity and inhibit the silicon oxide formation and corrosion.

(page 10, line No:290-297)

 Overall, it is evident that rGO as the co-catalyst of the p-Si photocathode is capable of encouraging the reaction kinetics between the interfacial layers of the buried electrode and electrolyte and acts as the passivation layer that preserves the p-Si photocathode from corrosion and native oxide formation. It is obvious from the gained results, in rGO deposited p-Si, the minority carriers generated by the optical absorption can move with no trouble and participate in the evolution of hydrogen more rapidly than in the bare p-Si electrode. Therefore, rGO optimizes negative effects of the interfacial layer between the p-Si and electrolyte in an integrated fashion to harvest more hydrogen, making it an excellent baseline co-catalyst to investigate in PEC cell.

Reviewer 3 Report

This work presents that both graphene oxide (GO) and reduced-graphene oxide (r-GO) can be prepared using argon gas plasma and deposited onto Si to enhance the photoelectrochemical hydrogen evolution of Si. The background of the manuscript clearly points out the necessity of a passivation layer on Si but the transition from this to GO is lacking. Authors are suggested to add that in the introduction section. With extensive characterizations performed, the conclusion is persuasive with various data. I would like to recommend this manuscript to be accepted after authors address some of my concerns. 

Authors used many abbreviations but without explaining the meaning, which makes the manuscript sometimes hard to understand.  Authors should pay attention to the superscript and subscript, especially for the expression of units and H2.  Line 72-74 is hard to understand. Please rewrite.  Sometimes GO and r-GO are indistinguishable according to the writing. Please specify. For example, between line 111-126.  The units are not place properly. For example, if several items share the same unit, please only put one unit in the end.  I don't understand why GO after treatment of argon plasma will gain N based on XPS measurement? In line 173, these are functional groups instead of compounds.  References are a bit old.  I was wondering if TGA can help this work to illustrate the successful deposition. 

Author Response

Title: In-situ Deposition of Graphene Oxide Catalyst for Efficient Photoelectrochemical Hydrogen Evolution Reaction using Atmospheric Plasma

Authors: Khurshed Alam, Yelyn Sim, Ji-Hun Yu, Janani Gnanaprakasam, Hyeonuk Choi, Yujin Chae, Uk Sim*, and Hoonsung Cho*

We appreciate the editors and referees for providing the valuable comments and suggestions to improve the quality of our recent submission for the consideration of publication by Applied Surface Science. According to the reviewers’ comments, we have thoroughly revised this manuscript. The corrections are highlighted in red in the main manuscript. The followings are our point-by-point responses:

Thank you for your kind consideration.

Response to Reviewers’ Comments

Reviewer: 3

This work presents that both graphene oxide (GO) and reduced-graphene oxide (r-GO) can be prepared using argon gas plasma and deposited onto Si to enhance the photoelectrochemical hydrogen evolution of Si. The background of the manuscript clearly points out the necessity of a passivation layer on Si but the transition from this to GO is lacking. Authors are suggested to add that in the introduction section. With extensive characterizations performed, the conclusion is persuasive with various data. I would like to recommend this manuscript to be accepted after authors address some of my concerns.

Authors used many abbreviations but without explaining the meaning, which makes the manuscript sometimes hard to understand. Authors should pay attention to the superscript and subscript, especially for the expression of units and H2. Line 72-74 is hard to understand. Please rewrite. Sometimes GO and r-GO are indistinguishable according to the writing. Please specify. For example, between line 111-126.  The units are not place properly. For example, if several items share the same unit, please only put one unit in the end. I don't understand why GO after treatment of argon plasma will gain N based on XPS measurement? In line 173, these are functional groups instead of compounds. References are a bit old. I was wondering if TGA can help this work to illustrate the successful deposition.

Response:

â–  Thank you for pointing out these crucial mistakes. In response to your comment, we have corrected unit in manuscript.

(page 1, line No:43)

Photoelectrochemical cell (PEC) which can produce H2 production by solar energy-driven photocatalysis is the effectual device for the production of hydrogen.

â–  We rewrite ‘Line 72-74’ more clearly to help understand the purpose of this paper.

(page 3, line No:71-74)

In this study, we suggest new pathway to direct deposition of GO and rGO onto Si substrates method and the prepared hydrophobic rGO and hydrophilic GO are applied as a PEC co-catalysts for hydrogen production to confirm the correlation between the surface property and catalytic performance.

â–  Furthermore, we have revised our manuscript to correct ambiguity regarding terminology.

(page 7-8, line No:215-226)
On the basis of the HER onset potential values hydrophobic rGO started progressing earlier than hydrophilic GO, and bare p-Si. Followed by the onset potential, the current density increases gradually with the negative applied potential. The current density of p-Si wafer is -0.4 mA cm-2. It was further increased by the coated rGO on p-Si wafer, hydrophobic rGO of -0.15 V and hydrophilic GO of -0.25 V respectively. It hints that the problematic p-Si wafer behaved healthier with rGO as the co-catalyst and also passivating layer by condensing the photo-oxidation and photo-corrosion.

(page 8, line No:234-238)

Differing from the current density loss in p-Si wafer, the current density of the hydrophilic GO (-0.33 V vs. RHE) and hydrophobic rGO (-0.1 V vs. RHE) is improved. From this outcome, it is clear that the incorporated rGO on p-Si wafer as an effective layer would enhance the activity and inhibit the silicon oxide formation and corrosion.

(page 10, line No:290-297)

 Overall, it is evident that rGO as the co-catalyst of the p-Si photocathode is capable of encouraging the reaction kinetics between the interfacial layers of the buried electrode and electrolyte and acts as the passivation layer that preserves the p-Si photocathode from corrosion and native oxide formation. It is obvious from the gained results, in rGO deposited p-Si, the minority carriers generated by the optical absorption can move with no trouble and participate in the evolution of hydrogen more rapidly than in the bare p-Si electrode. Therefore, rGO optimizes negative effects of the interfacial layer between the p-Si and electrolyte in an integrated fashion to harvest more hydrogen, making it an excellent baseline co-catalyst to investigate in PEC cell.

(page 3, line No:111-123)

Chemical composition analysis was carried out by the Raman spectroscopic and X-ray photoelectron spectroscopic technique. From the Raman spectroscopy results, it was confirmed that the as-deposited rGO and GO were distinguishable from graphene. The typical peaks for graphene are D (1350 cm-1), G (1580 cm-1) and 2D peak at 2690 cm-1. Pristine graphene does not show any D peak which represents edges of a graphene crystal and chemical bonds [12], while typical Raman spectra for GO are characterized by its D and G band corresponding to 1353 cm-1 and 1605 cm-1 respectively [13]. Raman spectrum in Fig. 1 (a) shows D, G and 2D peaks at 1345 cm-1, 1585 cm-1 and 2685 cm-1 respectively which indicates that rGO was successfully deposited. A slight redshift in G peak due to multi-layer deposition upon one another and by defects such as vacancies, grain boundaries [14-17]. The intensities of D, G and 2D peaks for Fig. 1 (a) are 351, 311 and 268 respectively and the intensity ratio of I2D/IG is 0.86 which shows multi-layered reduced graphene oxide [18]. Raman spectrum in Fig. 1 (b) shows D, G and 2D peaks at 1352 cm-1, and 1593 cm-1 and 2689 cm-1 respectively. The shifted peaks compared to the peaks in Fig. 1 (a) matched closely to the peak in hydrophilic GO [13]. The intensities of D, G and 2D peaks for Fig. 1 (b) are 460, 358, and 300 respectively and the ratio of I2D/IG is 0.83 indicated multi-layered deposition of graphene oxide [18]. The intensity of D peak in hydrophilic GO is attributed to substantial functionalization of GO, which is confirmed by XPS analysis [19].

â–  The reason that why why GO after treatment of argon plasma will gain N based on XPS measurement is that; the hydrophobic and hydrophilic characters are independent of time. We have observed that transition from hydrophobic to hydrophilic characters can only be obtained by argon plasma treatment alone while turning carbon source gas off. The reason behind transition from hydrophobic to hydrophilic is that energy from argon plasma is transferred to deposited rGO by radiation keeping distance of 1 cm, as a result transferring energy is breaking sp2 carbon atoms, allowing oxygen atoms from the environment to react on the surface of carbon atoms that makes it hydrophilic as well as minute amount of nitrogen atoms are also added from the environment as our process is operated in an open environment.

â–  With respect to your correction of ‘In line 173, these are functional groups instead of compounds’, we correct our sentence.

(page 6, line No:175)

Functional groups like C-OH, C-H, C-O-C, -COOH, C-O-C, C-OH, –C=O, C=O, and carboxylates (O-C=O) make hydrophilic GO surface due to dynamic interaction with water by generating intermolecular forces called hydrogen bonding

â–  Furthermore, we added latest references which have been published within 2 years.

De Marchi, L.; Pretti, C.; Gabriel, B.; Marques, P. A.; Freitas, R.; Neto, V., An overview of graphene materials: Properties, applications and toxicity on aquatic environments. Science of the Total Environment 2018, 631, 1440-1456. Askari, M. B.; Salarizadeh, P.; Rozati, S. M.; Seifi, M., Two-dimensional transition metal chalcogenide composite/reduced graphene oxide hybrid materials for hydrogen evolution application. Polyhedron 2019, 162, 201-206. Zhang, Q.; Li, T.; Luo, J.; Liu, B.; Liang, J.; Wang, N.; Kong, X.; Li, B.; Wei, C.; Zhao, Y., Ti/Co-S catalyst covered amorphous Si-based photocathodes with high photovoltage for the HER in non-acid environments. Journal of Materials Chemistry A 2018, 6, (3), 811-816.

Round 2

Reviewer 2 Report

The manuscript has been satisfactorily revised.

(1) XPS results:

> We have performed XPS analysis using titanium substrate as a substrate.

This should be described in the manuscript and shown in the Figure.

(2) Photocurrent density - potential curve:

> The values were measured and extrapolated by our group referring to the figure 5.

Reliability of the extrapolation is still ambiguous because the relationship between potential and current density is not linear. Method for the extrapolation should be briefly explained.

(3) Method:

> The X-ray source we used for analysis is aluminum (Al).

This should be described in the Method section.

(4) Spelling etc.:

Some errors in spelling and letter usage still remain. Please check again.

For example, 

• Methane --> methane 

• Argon --> argon

• science (journal name) --> Science    etc.

Author Response

Title: In-situ Deposition of Graphene Oxide Catalyst for Efficient Photoelectrochemical Hydrogen Evolution Reaction using Atmospheric Plasma

Authors: Khurshed Alam, Yelyn Sim, Ji-Hun Yu, Janani Gnanaprakasam, Hyeonuk Choi, Yujin Chae, Uk Sim*, and Hoonsung Cho*

We appreciate the editors and referees for providing the valuable comments and suggestions to improve the quality of our recent submission for the consideration of publication by Applied Surface Science. According to the reviewers’ comments, we have thoroughly revised this manuscript. The corrections are highlighted in red in the main manuscript. The followings are our point-by-point responses:

Thank you for your kind consideration.

Response to Reviewers’ Comments

Reviewer 2:

1) XPS results:

> We have performed XPS analysis using titanium as a substrate.

This should be described in the manuscript and shown in the Figure.

Response:

Thank you for your comments. As the reviewer stated, XPS analysis didn’t describe titanium peak. We updated XPS analysis inserted titanium peak in Figure 2a and 3a. Moreover, we mentioned about the titanium substrate in a place where XPS description is given.

(page 3, line No:97)

High-Performance X-ray Photoelectron Spectroscopy (XPS, Model: K-ALPHA +, Source aluminum (Al)) with Kα X-ray analyzer was performed to study the various functional groups of hydrophobic rGO and hydrophilic GO deposited on titanium substrate.

(page 4, line No:130-131)

Figure. 2. The XPS spectra survey of the as-deposited hydrophobic rGO. (a) XPS wide-scan spectrum and (b) XPS high-resolution spectrum registered for the energy range of C1s and (c) O1s.

(page 5, line No:144-145)

Figure. 3. The XPS spectra survey of the hydrophilic GO. (a) XPS wide-scan spectrum, (b) XPS high-resolution spectrum registered for the energy range of C1s,  (c) O1s, and (d) N1s spectra of GO.

(2) Photocurrent density - potential curve:

> The values were measured and extrapolated by our group referring to the figure 5.

 Reliability of the extrapolation is still ambiguous because the relationship between potential and current density is not linear. Method for the extrapolation should be briefly explained.

Response:

Thank you for precious comment. In this work, photocurrent density – potential curves obscured saturation current density, because we measured under the potential from 0.0 V to -0.6 V vs. RHE to find the starting point of the current increment as shown in Figure 5 (a). Then, we showed the saturation current density, -3.5 mA cm-2 of hydrophobic rGO in Figure 5 (b). We calculated the onset potential of bare p-Si by extrapolating from -0.5 V to -0.6 V vs. RHE with the saturation current density as -3.5 mA cm-2 shown in figure5 (b).

(page 8-9, line No:247-259)

At -0.1mA cm-2 there is betterment in photovoltage of hydrophobic rGO (0.52 V) in relation with the hydrophilic GO (0.3V of photovoltage) and the bare p-Si electrode (0.16V of photovoltage) (accurate vales are descried in Table 2) (The extrapolated value is calculated through a linear sweep potential from -0.5 V to -0.6 V, and the linear graph assumes saturated at a saturation current density of -3.5 mA cm-2 shown in figure5 (b)).

(3) Method:

> We used aluminum (Al) as the X-ray source.

 This should be described in the Method section.

Response:

             Thank you for your comments. As the reviewer stated, we mentioned about source of the X-ray, aluminum (Al) in method section 2.2.

(page 2, line No:90-92)

High-Performance X-ray Photoelectron Spectroscopy (XPS, Model: K-ALPHA +, Source aluminum (Al)) with Kα X-ray analyzer was performed to study the various functional groups of hydrophobic rGO and hydrophilic

(4) Spelling etc.:

Some errors in spelling and letter usage still remain. Please check again.

Response:

We apologize for those minor mistakes. In response to your comment, we have revised our manuscript to correct spelling.

(page 2, line No:59-60)

Generally, for the deposition of GO onto diverse substrates directly, pricey vacuum plasma has been utilized in the deposition technique, rather than atmospheric plasma.

(page 2, line No:61-62)

it is evitable using copper as a substrate to grow the N-doped graphene during chemical vapor deposition

(page 2, line No:66-68)
A mixture of methane and Ar gas was streamed to stimulate the formation of bare hydrophobic rGO, while the preparation of hydrophilic GO was inspired under additional Ar plasma treatment.

(page 2, line No:81-82)
The plasma was generated with the power of 240W (900 MHz) using argon gas at a rate of 4 l/m as a plasma carrier.

(page 2, line No:82-83)
A mixture of methane (10%) and argon gas was used and introduced to the plasma at a rate of 60 SCCM.

(page 2, line No:84-85)

The GO was obtained by the additional treatment of plasma with argon gas as the carbon source for one min.

(page 3, line No:108-109)

The 300 W Xe lamp with a light intensity of 100 mW cm-2 using a glass Air Mass 1.5 Global filter was used as the source of visible light for radiance on the substrate.

(page 7, line No:197-200)

(a) & (b) Photocurrent density – potential curves of bare p-Si, hydrophilic GO and hydrophobic rGO under dark and light conditions and the potential swept from 0 V to – 0.6 V vs. RHE in a three-electrode cell. (c) Current increment between dark and light conditions (d) Photovoltage of bare p-Si, hydrophilic GO and hydrophobic rGO at -0.1 mA cm-2.

(page 7, line No:207-209)

Firstly, current density, the measurement of electric current flowing per-unit cross-sectional area of a material is measured in 1 M of aqueous perchloric acid (HClO4) under acid solution (pH = 0) as the potential stroked from 0.0 V to -0.6 V vs. RHE in a three-electrode cell.

(page 7, line No:213-214)

The onset potentials, which represents the starting point of the current increment (-0.1mA cm-2) were observed for all the three samples.

(page 7, line No:214-215)

The observed onset potential values, which indicates current density of -0.1 mA cm-2 under dark condition are as follows,

(page 8, line No:224)

The current density of p-Si wafer is -0.4 mA cm-2.

(page 8, line No:219-220)

Table 2. Summary of the experimental data of bare p-Si, hydrophilic GO and hydrophobic rGO as co-catalyst on Si electrode.

Condition

Bare p-Si

Hydrophilic
GO

Hydrophobic
rGO

Onset potential at dark condition (V)

-0.85†

-0.66

-0.62

Potential at -0.1mA/cm2 at light condition (V)

-0.69†

-0.33

-0.1

Photovoltage* (V)

0.16

0.3

0.52

Current increment at specific potential between dark and light condition (mA/cm2)

at -0.2VRHE

0.003

0.026

0.498

at -0.4VRHE

0.007

0.251

2.991

at -0.6VRHE

0.018

2.1

3.353

(page 9, line No:262-264)

Impedance measurements were performed to study the resistance prevails in the charge transfer of hydrophobic rGO and hydrophilic GO on p-Si wafer with a frequency of 106 – 1 Hz, an amplitude of 5 mV, and at 0 V vs. RHE in a three-electrode system in Fig. 6 (a).

(page 11, line No:324-325)
Geim, A. K., Graphene: status and prospects. Science 2009, 324, (5934), 1530-1534.

(page 11, line No:326-328)

Zhang, Y.; Zhang, L.; Zhou, C., Review of chemical vapor deposition of graphene and related applications. Accounts of Chemical Research 2013, 46, (10), 2329-2339.

(page 11, line No:342-343)

Dunn, S., Hydrogen futures: toward a sustainable energy system. International Journal of Hydrogen Energy 2002, 27, (3), 235-264.

(page 11, line No:344-348)

Ji, L.; McDaniel, M. D.; Wang, S.; Posadas, A. B.; Li, X.; Huang, H.; Lee, J. C.; Demkov, A. A.; Bard, A. J.; Ekerdt, J. G., A silicon-based photocathode for water reduction with an epitaxial SrTiO 3 protection layer and a nanostructured catalyst. Nature Nanotechnology 2015, 10, (1), 84.

(page 12, line No:377-380)

Childres, I., Jauregui, Luis A,Park , Wonjun, Cao, Helin and Chen, Yong P, <1.RAMAN SPECTROSCOPY OF GRAPHENE AND RELATED MATERIAL.pdf>. New Developments in Photon and Materials Research 2013, 1.

(page 12, line No:381-384)

Ferrari, A. C.; Meyer, J.; Scardaci, V.; Casiraghi, C.; Lazzeri, M.; Mauri, F.; Piscanec, S.; Jiang, D.; Novoselov, K.; Roth, S., Raman spectrum of graphene and graphene layers. Physical Review Letters 2006, 97, (18), 187401.

(page 12, line No:392-394)

Shen, Y.; Lua, A. C., A facile method for the large-scale continuous synthesis of graphene sheets using a novel catalyst. Scientific Reports 2013, 3.

(page 12, line No:399-401)

Fan, X.; Zhang, G.; Zhang, F., Multiple roles of graphene in heterogeneous catalysis. Chemical Society Reviews 2015, 44, (10), 3023-3035.

(page 13, line No:412-415)

Stankovich, S.; Dikin, D. A.; Piner, R. D.; Kohlhaas, K. A.; Kleinhammes, A.; Jia, Y.; Wu, Y.; Nguyen, S. T.; Ruoff, R. S., Synthesis of graphene-based nanosheets via chemical reduction of exfoliated graphite oxide. Carbon 2007, 45, (7), 1558-1565.

(page 13, line No:419-422)

Depan, D. a. B. a. J. S. b. R. D. K. a., Structure–process–property relationship of the polar graphene oxide-mediated cellular response and stimulated growth of osteoblasts on hybrid chitosan network structure nanocomposite scaffolds. Acta Biomaterialia 2011, 7, (9), 3432.
